# A 3.7-to-10 GHz Low Phase Noise Wideband LC-VCO Array in 55-nm CMOS Technology

**Yan Yao**  **, Zhiqun Li \*, Zhennan Li, Bofan Chen and Xiaowei Wang**

Institute of RF-& OE-ICs, Engineering Research Center of RF-ICs and RF-Systems, Ministry of Education, Southeast University, Nanjing 210096, China; 230179055@seu.edu.cn (Y.Y.); 230189057@seu.edu.cn (Z.L.); chenbofan@seu.edu.cn (B.C.); 230198961@seu.edu.cn (X.W.)
\* Correspondence: zhiqunli@seu.edu.cn

**Abstract:** This paper presents a four-core LC-VCO array in 55 nm CMOS technology. Based on the multi-core VCO array technology and the switched capacitor array technology, the tuning range is expanded, and the phase noise optimization in a wide tuning range is achieved based on the second harmonic noise filtering technology and the Q value degeneration technology, as well as the optimization of the capacitor array switching transistors. The proposed VCO array, occupying a chip area of $1.65 \times 1.44$ mm$^2$, realizes a measured oscillation frequency range of about $3.7-10$ GHz with phase noise of $-127.5 \sim -116.08$ dBc/Hz at 1 MHz frequency offset, and achieves an output power of 2.69 dBm from a total power consumption of 52.8 mW.

**Keywords:** VCO array; low phase noise; wideband; CMOS; SDR



## 1. Introduction

As the core module of wideband frequency synthesizer based on phase locked loop (PLL) system for software-defined radio (SDR) application [1,2], the design and optimization of wideband voltage-controlled oscillator (VCO) is of great importance. Firstly, the tuning range and phase noise performance of VCO directly determine the output frequency range and out-of-band phase noise performance of PLL system. Secondly, the power consumption and chip area of VCO usually occupy most of the power consumption and area of PLL system. Finally, the degree of VCO's suppression of common-mode noise near oscillation frequency is directly related to the spurious performance of PLL.

At present, the main research directions of VCO are still focused on the improvement of phase noise performance [3–15], the expansion of tuning range [16–21], and the reduction of power consumption [5–7,14,19,22,23]. There are also some other aspects of optimization, such as tuning gain flatness and output swing enhancement [23–25]. Common methods to expand the tuning range include the use of switched capacitor arrays with variable capacitors, and the use of switched tunable inductors. Common methods to reduce phase noise include the use of current source filtering to eliminate current source noise, and the use of differential control structures to reduce control signal noise. The increase in the number of devices means the increase in noise, so there is usually a compromise between tuning range expansion and phase noise optimization, and simultaneous optimization of tuning range and phase noise performance is rare. How to reduce the phase noise of the VCO under the premise of ensuring a wide tuning range, and how to achieve the optimization of the VCO phase noise in the full tuning range has been little studied. However, for SDR applications, it is extremely important to consider both VCO tuning range expansion and phase noise optimization. In this letter, the tuning range expansion technology combined with multi-core VCO array and switched capacitor array and phase noise optimization strategy of wideband second harmonic noise filtering technology based on Q value degeneration are explored comprehensively. Based on the parameter optimization of VCO core components, the optimization design of a 3.7–10 GHz VCO array is finally completed,

which simultaneously takes into account the requirements of low phase noise and wide tuning range.

## 2. VCO Array Architecture

The wide frequency tuning range can be acquired by combining the multi-core VCO array technology with the on-core switched capacitor array technology, and then the phase noise optimization for each VCO core based on a variety of technologies can be taken into account to improve the performance of phase noise. The block diagram of the proposed VCO array is shown in Figure 1, including four VCO cores, all of which are realized by NMOS cross-coupled LC oscillator structure. Each VCO core corresponds to a working sub-band, which is achieved by adjusting the parameters of the resonant tank. A certain redundant overlapping area is set between adjacent sub-bands to prevent the occurrence of frequency blind spots. The resonant tank of each VCO core adds a three-bit switched capacitor array for coarse frequency adjustment on the basis of the variable capacitor, with a total of eight tuning curves, which further expands the tuning range of the VCO core and maintains a low tuning gain to ensure the phase noise performance.

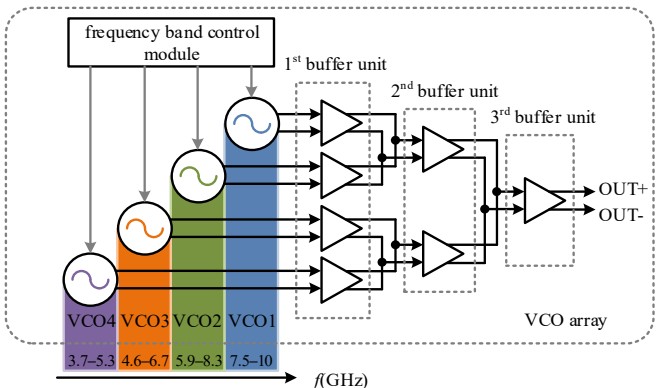

**Figure 1.** Block diagram of the proposed VCO array.

In addition, the VCO array also includes a three-stage buffer unit and a frequency band control module. The three-stage buffer unit is used to provide sufficient output signal amplitude for subsequent circuit. The second-stage buffer unit of VCO1 and VCO2 is shared, the second-stage buffer unit of VCO3 and VCO4 is shared, and the third-stage buffer unit is shared by VCO1, VCO2, VCO3, and VCO4. The buffer units are all implemented based on inverters with a wide operating bandwidth, where both the first and second stage buffer units are implemented by a one-stage inverter and the third stage buffer unit is cascaded by a three-stage inverter to enhance its driving capability. The frequency band control module is a serial-parallel conversion circuit, of which the output signal controls the working state of each VCO core by controlling the on-off of the switching transistor in the tail current control block of each VCO core, ensuring that only a single VCO core is working at the same time. In this state, the rest of the VCO cores do not work, so that mutual interference between signals in various frequency bands can be prevented and power consumption can be saved. All VCO cores and buffer units adopt differential structure, which can effectively suppress noise interference from power supply and external environment.

## 3. Circuit Design

Figure 2 presents the core circuit schematic of the NMOS cross-coupled LC oscillator used in this paper, whose oscillation frequency is determined by the three-bit switched capacitor control word $SW_{2\text{-}0}$ together with the tuning voltage $V_{\text{tune}}$, and the working current can be controlled by the current control word $B_{2\text{-}0}$ through the three-bit switched resistance at the ground terminal. When the three-bit switched resistors are disabled, the corresponding VCO core does not operate, so that the four VCO cores in the VCO array

can be switched. The switched capacitor control word and tuning voltage of the four VCO cores are shared.

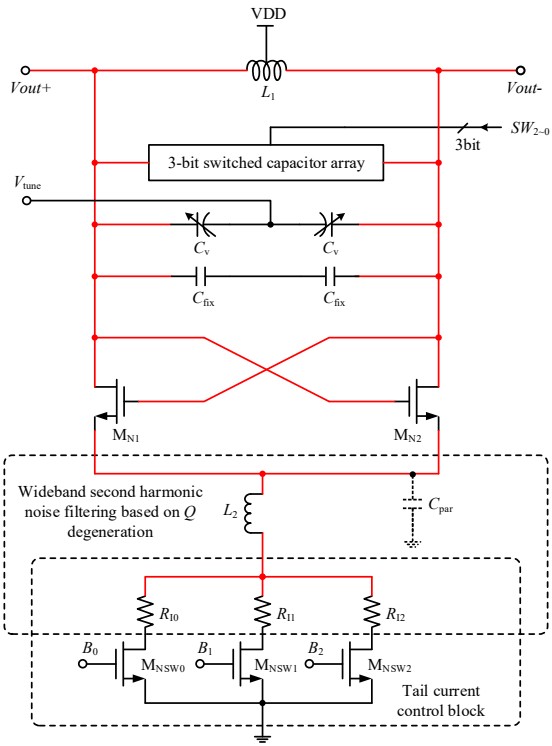

**Figure 2.** Schematic of the NMOS cross-coupled LC oscillator.

It is well known that the second harmonic noise filtering technique can achieve the optimization of VCO phase noise, but due to the bandwidth limitation of the filter, this technique usually has a limited frequency range that can be optimized, and can only achieve a better optimization effect in a narrow tuning range corresponding to the resonant frequency of the inserted inductor and the parasitic capacitor. To achieve the optimization of VCO phase noise across a wide frequency tuning range, a phase noise optimization technique based on second harmonic noise filtering and Q value degeneration is proposed in this design. As shown in Figure 2, an inductor is connected in series between the common source of the differential transistor pair and the tail current control block. The inductor is selected to resonate with the total capacitance seen from the common source of the differential transistor pair at a frequency twice the VCO oscillation frequency, thus forming a second harmonic filter. The advantage of this structure is that the resistor in series with the inductor in the tail current control block reduces the Q value of the second harmonic filter, thus extending its operating bandwidth, allowing the second harmonic filter technique to function over a wider frequency range and optimizing phase noise performance over the whole VCO tuning range. Figure 3 presents the simulated phase noise curves of the single VCO core with the second harmonic noise filtering technique at the lowest and highest resonant frequencies with and without the tail current control block. As indicated in the figure, without the tail current control block, the phase noise of the single VCO core at 1 MHz frequency offset varies from −134.11 dBc/Hz to −121.76 dBc/Hz, which fluctuates more than 12 dB; while with the tail current control block, the phase noise of the single VCO core at 1 MHz frequency offset varies from −133.44 dBc/Hz to −129.82 dBc/Hz, with a fluctuation of about 3.6 dB. It can be seen that the second harmonic noise filtering technique combined with the tail current control block can optimize the phase noise over a wider tuning range with almost no impact on the optimal optimization effect and with reduced power consumption. In addition, the tail current control block allows adjustment of the operating current to suit different process corner variations.

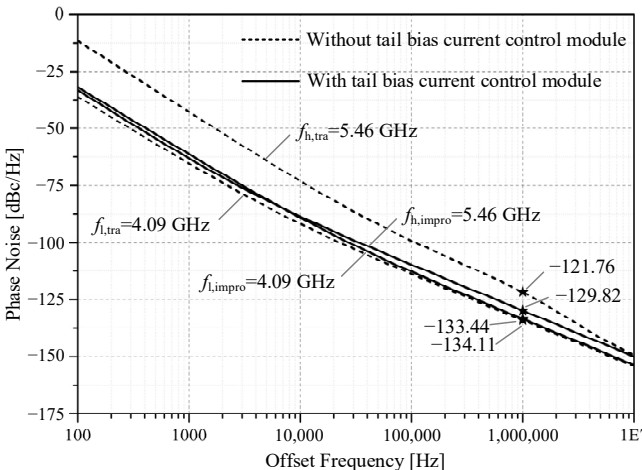

**Figure 3.** Simulated phase noise curves of the single VCO core employing the second harmonic noise filtering technique at the lowest and highest resonant frequencies with and without the tail current control block.

### 3.1. Inductor Selection

Due to the general high quality factor (Q value) of the capacitors, the Q value of the resonant tank is mainly determined by the Q value of the inductor. And the higher the Q value of the resonant tank, the lower the phase noise of the VCO. Therefore, selecting an inductor with a larger Q value contributes to the reduction of VCO phase noise. Generally speaking, for inductors with the same inductance, the larger the inner diameter, the higher the Q value. As for the inductor in the second harmonic filter, the inductor with a lower Q value should be selected to widen the filter bandwidth.

The selection of the inductance value $L$ of the resonant inductor should take into account the influence of $L$ on the tuning range and the influence of its equivalent parallel resistance on the oscillator start-up condition. On the one hand, the equivalent parallel resistance $R_p$ of the resonant inductor with oscillation frequency $f$ can be expressed as

$$R_p = 2\pi f \cdot Q_L L \tag{1}$$

It can be seen that when $f$ remains unchanged, the larger the product $Q_L L$ of the inductance value and the Q value is, the larger $R_p$ is, which is more conducive to the start-up of the oscillator. On the other hand, the oscillation frequency $f$ can be expressed as

$$f = \frac{1}{2\pi\sqrt{LC}} \tag{2}$$

At a fixed oscillation frequency, the increase of the inductance means the reduction of the total capacitance of the resonant tank, which also means that the variable range of the capacitance is narrowed, which will result in a reduced tuning range for the VCO. Therefore, the inductance value $L$ should not be too large.

### 3.2. Variable Capacitor Unit

As shown in Figure 2, the variable capacitor unit is composed of two accumulating MOS varactors $C_v$ connected in series with an intermediate tuning voltage $V_{tune}$, of which the variation range is 0~2.5 V. When the capacitance values of the varactors are the same, the larger the number of fingers of the varactor is, the larger the Q value is; the larger the $W/L$ is, the larger the Q value is, but the capacitance variation range will be slightly narrowed. Therefore, the size of the varactor should be selected in this direction.

### 3.3. Switched Capacitor Array

The schematic of the three-bit switched capacitor array for coarse tuning is shown in Figure 4. Each bit switched capacitor is composed of two identical fixed capacitors $C_{j(j=0,1,2)}$ connected in series with a switching transistor $M_{sj(j=0,1,2)}$, and the fixed capacitance is selected in the ratio of 1:2:4. When the control signal $SW_{j(j=0,1,2)}$ is high, $M_{sj}$ is turned on and can be equivalent to an on-resistance; when $SW_j$ is low, $M_{sj}$ is turned off and can be equivalent to an infinite resistance, so that the fixed capacitors are disconnected to the resonant tank. In order to reduce the on-resistance of the transistor to weaken the deterioration of the Q value of the resonant tank, it is necessary to increase the $W/L$ ratio of the transistor. But it will cause the parasitic capacitance of the transistor to become larger, which will affect the tuning range. So, there are tradeoffs in the design.

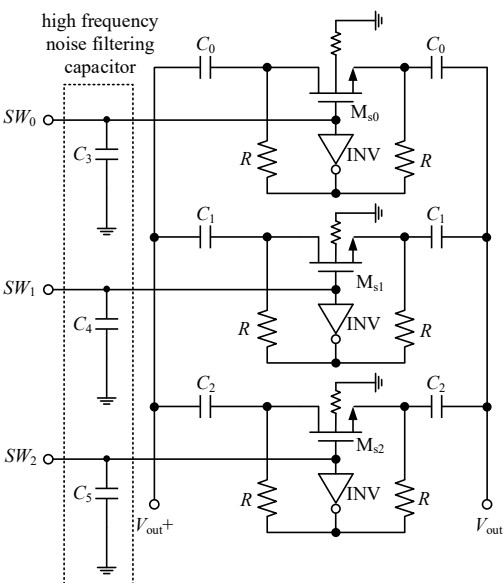

**Figure 4.** Schematic of the three-bit switched capacitor array.

The gate of the switching transistor is connected to the source and drain by an inverter and two resistors $R$. The inverter is mainly used to ensure that the transistor is completely turned on or off. The value of $R$ is important when $SW_j$ is low, as the connected switched capacitors need to be able to be treated as floating. The larger the $R$, the higher the Q value of the resonant tank and the better phase noise performance the VCO can achieve. However, when the value of $R$ reaches a certain level, the effect on the phase noise becomes less pronounced, and it takes up too much chip area instead. The resistance value of 10 kΩ is finalized through simulation.

The Q value of the switched capacitor array is mostly affected by the switching transistors. The deep N-well MOS tube with a large resistor connecting the body end to the ground is employed in this design, thereby improving the switch isolation and reducing the deterioration of the Q value of the resonant tank by the switch. In addition, the gate of each transistor is connected in parallel with a capacitor to ground, providing effective suppression of high-frequency noise leakage from the source and drain of the transistor to the gate and reducing its effect on the gate control signal.

### 3.4. Resonant Tank Parameter Design

The maximum and minimum resonant frequencies of each VCO core can be expressed as

$$f_{\text{VCO}i,\max} = \frac{1}{2\pi\sqrt{L_{\text{VCO}i}\left(C_{\text{fix,VCO}i}+C_{\text{v,VCO}i}+C_{\text{par,VCO}i}\right)}}, i = 1,2,3,4$$

$$f_{\text{VCO}i,\min} = \frac{1}{2\pi\sqrt{L_{\text{VCO}i}\left(C_{\text{fix,VCO}i}+C_{\text{v,VCO}i}+C_{\text{par,VCO}i}+\sum_{j=0}^{2}C_{j,\text{VCO}i}\right)}}, i = 1,2,3,4 \tag{3}$$

where $L_{\text{VCO}i}$ is the inductance, $C_{\text{fix,VCO}i}$ is the fixed capacitance, $C_{\text{v,VCO}i}$ is the varactor capacitance, $C_{\text{par,VCO}i}$ is the parasitic capacitance, and $C_{j,\text{VCO}i}$ is the switching capacitance. The tuning gain $K_{\text{VCO}i}$ of each VCO core can be obtained by differentiating Equation (3) with $V_{\text{tune}}$, and can be expressed as

$$K_{\text{VCO}i} = -2\pi^2 f_{\text{VCO}i}^3 L_{\text{VCO}i}\frac{dC_{\text{v,VCO}i}}{dV_{\text{tune}}}, i = 1,2,3,4 \tag{4}$$

It can be seen that the tuning gain is positively correlated with the inductance, the rate of change of the variable capacitance relative to $V_{\text{tune}}$, and the cube of the oscillation frequency. In order to prevent the huge difference in the tuning gain of different VCO cores, the size of the inductor and the varactor should be adjusted synchronously according to the change of the oscillation frequency during the design.

## 4. Measurement Results

The proposed four-core VCO array is implemented in the SMIC 55 nm 1P10M CMOS technology. The micrograph of the chip is shown in Figure 5. The area of the chip, including all of the pads, is 2.37 mm$^2$. The PCB photo and test bench photo of the proposed four-core VCO array are presented in Figure 6.

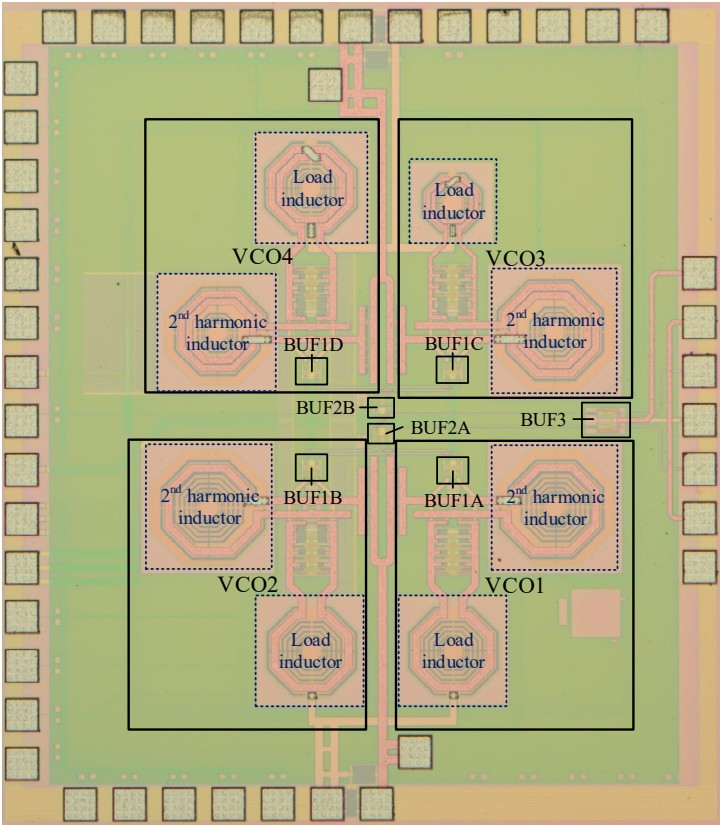

**Figure 5.** Chip micrograph of the proposed four-core VCO array.

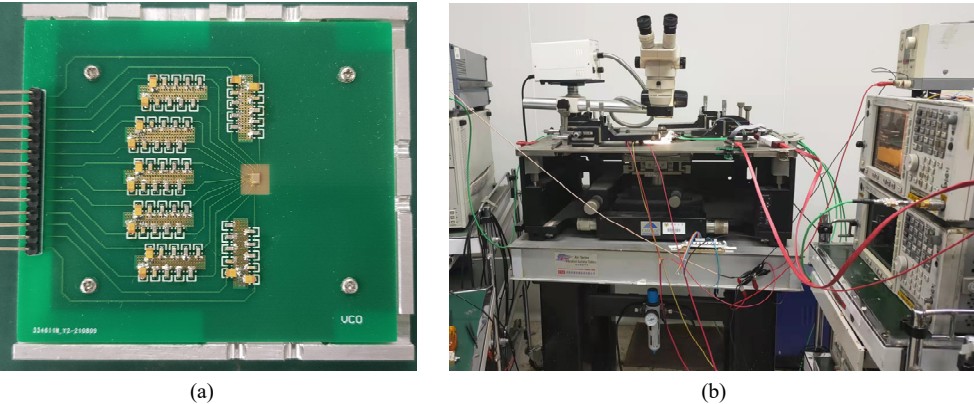

(a)　　　　　　　　　　　　　　　　　　　　(b)

**Figure 6.** (**a**) PCB photo (**b**) test bench photo of the proposed four-core VCO array.

In order to reduce the interaction between the RF signal routing and the bias and digital control signal routing, a large area of M6 layer metal only connecting the AC ground pad is laid on the entire layout, which can be used as a reference ground plane. This layer of large area metal is only used as a loop for AC signal, and no DC current flows, which can prevent the AC signal from being affected by external environment. The layers above M6, which have larger metal thickness and higher electrical conductivity, are suitable for RF signal routing, while M2-M5 layers are used for bias and digital control signal routing and M1 layer is used for guard rings to reduce substrate noise. The above wiring scheme also facilitates electromagnetic (EM) co-simulation. As shown in Figure 7, the Momentum tool in ADS software is used to conduct EM simulation only for RF signal routing susceptible to parasitic effect (the part marked in red in Figure 2), and then combining with post-simulation of remaining circuits, the circuit performance close to the actual situation can be obtained.

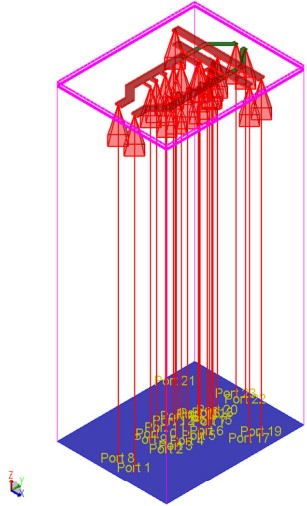

**Figure 7.** EM simulation for RF signal routing.

The measured frequency tuning curves for the four-core VCO array are presented in Figure 8. The frequency range can cover 3.66–9.95 GHz. Due to the conservative design, the redundant overlapping area of adjacent tuning curves is set too large, and the same is true for adjacent VCO cores, resulting in a waste of chip area. The total power consumption of the VCO array is 52.8 mW, of which the power consumption of the VCO cores is 33.2 mW under 1.2 V supply voltage, and the power consumption of the buffer is 19.6 mW under 1.8&1.2 V supply voltage. Figure 9a,b demonstrate the measured output spectrum and phase noise curve of the VCO array at about 4 GHz oscillation frequency, respectively. The output power is 2.69 dBm, and the phase noise is −126.54 dBc/Hz at 1 MHz frequency

offset from 4 GHz carrier. The phase noise measurement results of the VCO array at 100 kHz, 1 MHz and 10 MHz frequency offsets across full frequency tuning range are shown in Figure 10. As indicated in the figure, the phase noise at 1 MHz frequency offset ranges from −127.5 dBc/Hz to −116.08 dBc/Hz, with a fluctuation within 11.5 dB over the full frequency range. It can be seen that the phase noise fluctuation in the full frequency range of the designed VCO array is small, which is consistent with the simulation results and achieves the balance of tuning range and phase noise performance. Nevertheless, there is still a gap between the measurement results and simulation results of VCO phase noise, which is supposed to be due to insufficient consideration of EM simulation. In this design, in order to speed up the simulation, only the core RF signal routes of each VCO core were simulated separately, without considering the mutual influence of the routes between the four VCO cores and the mutual influence of the inductors, which may lead to the deviation of the second harmonic filtering range, thus leading to the second harmonic filtering technology not playing its best role. In the next step of VCO optimization design, the RF signal routs of all VCO cores together with the inductors should be considered as a whole for EM simulation to fully take into account their interactions. The performance of the VCO array is summarized and compared with current state-of-the-art VCOs in Table 1, which appears to be very competitive in terms of tuning range and phase noise metrics.

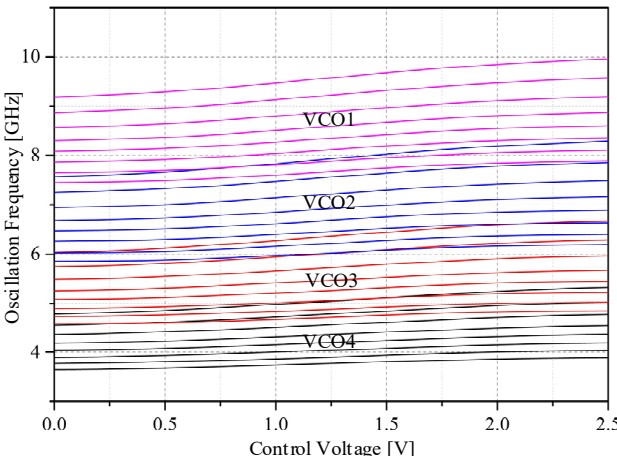

**Figure 8.** Measured frequency tuning curves.

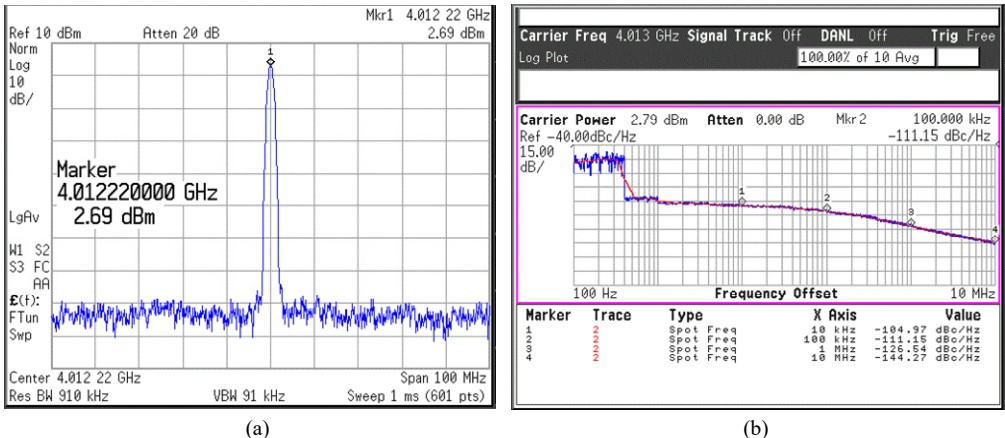

**Figure 9.** Measured (**a**) output spectrum and (**b**) phase noise curve at about 4 GHz oscillation frequency.

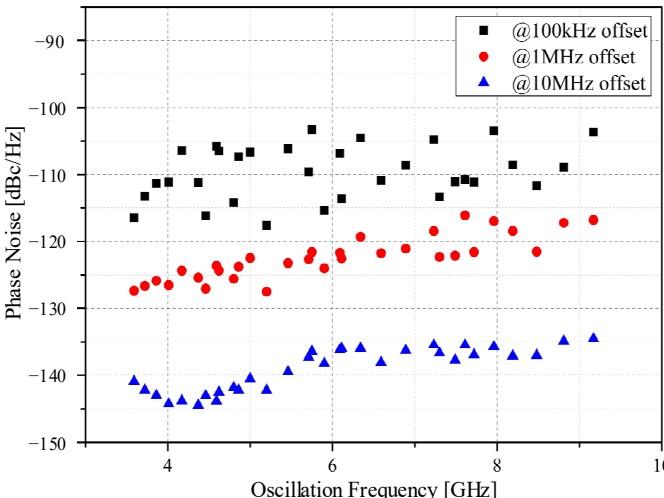

**Figure 10.** Measured phase noise at 100 kHz, 1 MHz, and 10 MHz frequency offsets across full frequency tuning range.

**Table 1.** Performance summary and comparison with current state-of-the-art VCOs.

| Ref | [14] | [16] | [17] | [22] | This Work |
|---|---|---|---|---|---|
| Technology | 180 nm CMOS | 28 nm CMOS | 130 nm CMOS | 65 nm CMOS | 55 nm CMOS |
| Supply voltage [V] | 0.78 | 1.8 | 1.2 | 0.6 | 1.2 |
| Frequency range [GHz] | 5.28–6.79 | 6.17–10.45 | 4.83–5.28 | 0.428–0.552 | 3.66–9.95 |
| Tuning range [%] | 25 | 51 | 8.9 | 25.2 | 92.4 |
| Phase noise [dBc/Hz @ 1 MHz] | −109.06 | −121.6~−115 | −109.5 | −94.84 | −127.5~−116.08 |
| Power consumption [mW] | 5.7 | 32–41 | 5.4 | 0.045 | 33.2 |
| FoM [1] [dB] | 177.3 | 182.4~179.3 | 177 | 162.1 | 180.8~183.5 |
| FoM_T [2] [dB] | – | 196.4~193.1 | 176 | – | 200.1~202.8 |
| Area [mm²] | 1 | 0.095 | 0.74 | – | 2.37 |

[1] FOM = $|PN| + 20\log_{10}(f_0/\Delta f) - 10\log_{10}(P_{DC}/1\mathrm{mW})$. [2] FOM_T = $\mathrm{FOM} + 20\log_{10}(TR/10)$.

## 5. Conclusions

This paper has demonstrated a LC-VCO array using various techniques simultaneously for frequency tuning range expansion and phase noise performance enhancement for SDR application. A multi-core and multi switched capacitor array technology is proposed for the expansion of the tuning range and a combined second harmonic noise filtering and Q value degeneration technology is proposed for the optimization of phase noise over a wide tuning range. Implemented in a 55 nm CMOS technology, the prototype design achieves phase noises of −127.5~−116.08 dBc/Hz at 1 MHz offset over 3.7−10 GHz frequency tuning range.

**Author Contributions:** Conceptualization, Y.Y. and Z.L. (Zhiqun Li); methodology, Y.Y.; software, Y.Y.; validation, Y.Y., Z.L. (Zhiqun Li), Z.L. (Zhennan Li), B.C., and X.W.; formal analysis, Y.Y.; investigation, Y.Y.; resources, Y.Y.; data curation, Y.Y.; writing—original draft preparation, Y.Y.; writing—review and editing, Y.Y.; visualization, Y.Y.; supervision, Z.L. (Zhiqun Li); project administration, Z.L. (Zhiqun Li); funding acquisition, Z.L. (Zhiqun Li). All authors have read and agreed to the published version of the manuscript.

**Funding:** This research was funded by National Key R&D Program of China, grant number 2019YFB2204602.

**Acknowledgments:** The authors would like to thank the SMIC for the fabrication.

**Conflicts of Interest:** The authors declare no conflict of interest.

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
