# Peer review of "A 3.7-to-10 GHz Low Phase Noise Wideband LC-VCO Array in 55-nm CMOS Technology"

_electronics, doi:10.3390/electronics11121897_

Round 1

Reviewer 1 Report

This paper presents a 3.7GHz ~ 10GHz LC-VCO array design with a wide frequency range using 4 VCOs.

The content and explanation of the paper are well done, but the originality that the paper is trying to claim is low. I think it would be better if you write a little more about what you want to claim.

1. Does the current indicated in the paper include the buffer current?

2. We recommend that you include measurement bench content for your measurements.

3. In Figure 5, it would be better to display the VCO's load inductor and 2nd harmonic inductor separately. Are vco3 and vco4 well marked?

4. It would be nice to have a picture showing the entire frequency tuning range at once.

5. Is there a reason why VCO1 has the smallest frequency range?

6. The buffer has a wide operating frequency range, so please add more about this.

Reviewer 2 Report

1)The background part should be riched with more information around the studied topic. What has been studied and what type of question is still unresolved.

2) For the Resutls part, the authros just list the data without any discussion or explaination, which is not good. Please provide some in-deep discussion around the measured data. 

3) Conclusions part should also be revised with the real conclusion information.

Round 2

Reviewer 1 Report

The paper has been well revised and edited.

Reviewer 2 Report

I am satisfied with the response